# The Relationship between Mood Symptom Severity and Perfectionism Subtypes in Mood Disorders: A Systematic Review and Meta-Analysis

**DOI:** 10.3390/brainsci13030377

**Published:** 2023-02-21

**Authors:** Katy Lea, Thomas Richardson, Nina Rauze

**Affiliations:** School of Psychology, University of Southampton Highfield Campus, University Road, Southampton SO17 1BJ, UK

**Keywords:** perfectionism, mood disorders, depression, bipolar disorder, depressive symptoms, manic symptoms

## Abstract

Background: Previous research suggests that there is a link between perfectionism and symptoms of depression. This study aimed to see if different types of perfectionism are linked differently to symptoms of depression in mood disorders and if there is a relationship between perfectionism and symptoms of mania in bipolar disorder. Methods: A systematic search was conducted in the databases PsycINFO, EMBASE, Web of Science, and PubMed to find papers which examined the relationship in clinical depression and bipolar disorder. A meta-analysis pooled the correlation effect sizes for mood symptoms severity and the severity of the perfectionism subtype. Results: Twelve papers were included in the review, with five of these being included in the meta-analysis. The meta-analysis found statistically significant positive correlations between greater severity of depression symptoms and more severe perfectionism for the following subtypes: concern over mistakes, doubts about actions, other-oriented perfectionism, parental criticism, self-oriented perfectionism, and socially prescribed perfectionism. There was no significant relationship between depression symptoms and perfectionism subtypes of organisation and personal standards. There were not enough studies reporting data for manic symptoms for the meta-analysis or for any firm conclusions to be drawn. Conclusions: The relationship between depression and perfectionism differs depending on the particular type of perfectionism examined. Most studies were cross-sectional and correlational, so causation cannot be inferred, and future longitudinal studies are needed.

## 1. Introduction

Depression is a mental illness characterised by symptoms such as low mood, hopelessness, poor concentration, and thoughts about death [1,2], and it is estimated to have a lifetime prevalence of 10.8% internationally [3]. Bipolar disorder is a mental illness characterised by episodes of depression with episodes of hypomania or mania characterised by symptoms including increased energy, pressure of speech, grandiose ideas, and an increase in goal-directed activity [1,2]. Bipolar disorder has a lifetime prevalence of 1.7% [4] and is linked to a number of negative outcomes, including increased suicide risk [5] and reduced life expectancy [6]. Depression globally is linked to 24.6 million years of living with disability per year [7] and substantial health costs [8]. Recent meta-analyses have demonstrated the effectiveness of psychological therapies for both bipolar disorder [9] and depression [10].

There is a large body of literature on the relationship between psychopathology and perfectionism [11]. Perfectionism involves holding extremely high standards and being self-critical when these standards are not met [11]. This can become maladaptive and negatively impact an individual’s life. The most comprehensive review and meta-analysis on the relationship between psychopathology and perfectionism to date, by Limburg et al. [11], found that most research investigating the relationship between psychopathological symptoms and perfectionism has been conducted in non-clinical populations (65.1%). This research found that higher levels of maladaptive perfectionism are related to higher levels of depressive symptoms [12]. However, it has been found that levels of perfectionism are higher in clinical populations than in non-clinical populations [13]. Therefore, clinical populations should be at the forefront of this research.

Limburg et al. [11] found that the three most evaluated psychological disorders in relation to perfectionism were anxiety disorders, eating disorders, and obsessive–compulsive disorder. This meta-analysis found positive correlations between various psychopathology outcomes and dimensions of perfectionism. Depressive symptoms were the most evaluated symptom in relation to perfectionism. However, this meta-analysis mainly looked at psychopathological symptoms over a range of disorders, and it only looked at these in relation to two overarching dimensions of perfectionism: perfectionistic strivings and perfectionistic concerns.

Maladaptive perfectionism has been found to predict self-reported depression in bipolar disorder [14]. Additionally, Scott et al. [15] found that perfectionism scores were a robust predictor of classification into either a bipolar disorder group or a control group. This suggests that perfectionism can predict the severity of mood disorders and, therefore, has high importance, as it may pose as an indicator of the risk of developing a mood disorder.

Other studies have shown that an increase in perfectionism is correlated with an increase in depression scores [16]. Hence, perfectionism appears to have a predictive relationship with depression in which the trait further contributes to the maintenance of depressive symptoms [17]. Furthermore, treatments targeting perfectionism may be effective in decreasing depressive symptoms [18]. Thus, expanding our understanding of the relationship between mood symptoms and perfectionism subtypes would be beneficial, as this will have implications for the development of effective clinical treatments. Therapies targeting the reduction of maladaptive perfectionism should be the focus of mood disorder treatment. 

Reviews of the literature have been conducted on the relationship between perfectionism and mood symptoms in some disorders. For example, a review on eating disorders found that those with eating disorders have higher scores on maladaptive perfectionism measures than those without [19]. A meta-analysis also found a link between perinatal mental and perfectionism, including depression specifically [20]. An earlier review found that perfectionism is higher in those with anxiety disorders, depression, and eating disorders than in those without, and that perfectionism may explain comorbidity across disorders [21].

However, there have been no meta-analyses focusing on the relationship between perfectionism and clinical mood disorders specifically and no meta-analyses on the relationship with bipolar disorder including manic symptoms. There have also been no meta-analyses which have examined whether different subtypes of perfectionism have a different relationship with depression. This study, therefore, aimed to conduct a systematic review and meta-analysis to examine the link between perfectionism subtypes and mood symptom severity in mood disorders.

## 2. Method

A systematic search was conducted across four databases in November 2021. These were PsycINFO, EMBASE, Web of Science, and PubMed. The following search terms were used: (“mood disorder*” OR “depressi* disorder” OR “affective disorder” OR “bipolar disorder”) AND (“perfect*”). In the first step, title and abstract searches were conducted. A citation search was completed in July 2022 to update the search, retrieving three additional papers. The papers that were accepted through title and abstract screening were then screened at the full paper level by two reviewers independently with 89% agreement. Discrepancies were discussed as a team and resolved.

The following inclusion criteria were used: papers had to include standardised measures of mood symptoms and perfectionism and report correlations between these measures. Samples had to include those with diagnosed mood disorders (depression and/or bipolar disorder), and participants had to be adults. The papers had to be in English in peer-reviewed journals.

A meta-analysis was performed using Comprehensive Meta-Analysis. This was conducted when two or more studies reported correlations for comparable measures of both mood symptom severity and measures of a perfectionism subtype or subscale of a perfectionism measure.

## 3. Results

### 3.1. Results of the Search

Figure 1 displays the flow diagram for the included studies. A total of 12 papers were included in the systematic review.

### 3.2. Study Characteristics

Seven of the included papers focussed on depressive disorders, three focussed on bipolar disorders, and two looked at both disorders. The design, participant characteristics, scales used, and key findings of each study are summarised in Table 1. Eleven of the included studies are cross-sectional designs using questionnaires to look at the relationship between mood symptoms in mood disorders and dimensions of perfectionism. One study was longitudinal. All looked at this relationship using correlational and/or regression analysis. Three of the studies were based in the United States, two were based in Canada, two were based in Australia, one was based in Germany, one was based in Belgium, one was based in Portugal, one was based in Turkey, and one was an international study. For participant recruitment, the majority of studies used outpatients (*n* = 8), one used participants from ongoing studies, one recruited participants via advertisements, one approached possible participants in medical centres, and one did not state their method of recruitment. Only three studies stated that the participants were stable in mood and on medication; the rest did not clarify whether participants were stable or unwell.

### 3.3. Measures Used

#### 3.3.1. Mood Symptom Measures

The most commonly used measures of depressive symptoms included the Beck Depression Inventory (BDI) [32], a 21-item self-report measure of depression symptoms with items rated from 0 to 3, for example, “I do not feel sad” (0) up to “I am so sad and unhappy all the time that I can’t stand it” (3); The Hamilton Depression Rating Scale (HDRS) [33] is a clinician-rated measure of 17 items rating the severity of depressive symptoms, such as work and activities (no difficulty = 0, stopped working = 4) and feelings of guilt (absent = 0, up to hears accusatory or denunciatory voices or threatening visual hallucinations = 4); The 21-Item Version of the Depression Anxiety Stress Scale (DASS-21) [34] depression subscale has seven self-reported items to measure symptom severity via statements such as “I couldn’t seem to experience any positive feeling at all” rated from 0 (did not apply to me at all) to 3 (applied to me very much or all of the time), and the Depressive Experiences Questionnaire [35] was also used; this is a self-report measure of experiences which are commonly seen in depression but not core depression symptoms, such as helplessness, guilt, and difficulties in relationships with family members. Other less frequently used measures were the Quick Inventory of Depressive Symptomatology—Self-Report [36], a 16-item self-report measure with questions about symptoms over the past week, such as falling asleep (never taking longer than 30 min to fall asleep up to talking more than 60 min most of the time); The Montgomery-Asberg Depression Rating Scale [37] is a clinician-rated scale to assess symptom severity of 10 items, such as difficulty concentrating and difficulty starting and completing activities; The Positive and Negative Affect Schedule [38] includes a 10-item scale on negative affect completed via self-report for the past week on a scale from not at all (1) to very much (5) for items such as “ashamed” and “interested”.

The most frequently used measure for manic symptoms was the Young Mania Rating Scale, an 11-item clinician-administered scale which assesses the severity of (hypo)manic symptoms, such as energy levels, grandiosity, sleep, and irritability, over the past 48 h [39]. The Internal State Scale (ISS), a 16-item self-report scale, was used to assess both depressive and manic symptoms with statements such as “Today I feel impulsive” and “Today I feel depressed”; these are rated from 1 (Not at all/Rarely) to 100 (Very much so/Much of the time) [40]. The General Behaviour Inventory (GBI), a 73-item self-report scale, was also used to assess both depressive and manic symptoms using questions such as “Has your mood or energy shifted rapidly back and forth from happy to sad or high to low?”, which are rated from 0 (Never or hardly ever) to 3 (Very often/Almost constantly) [41]. On all these measures, higher scores indicate more severe mood symptoms.

#### 3.3.2. Perfectionism Measures

The most used measures for perfectionism were the Frost Multidimensional Perfectionism Scale (including the Concern over Mistakes (CM), Doubting of Actions (DA), Parental Expectations (PE), Parental Criticism (PC), Organisation, and Personal Standards (PS) subscales) [42], the Hewitt and Flett Multidimensional Perfectionism Scale (including the Self-Oriented Perfectionism (SOP), Socially Prescribed Perfectionism (SPP), and Other-Oriented Perfectionism (OOP) subscales) [43], and the Dysfunctional Attitudes Scale Perfectionism subscale (DAS-P) [44]. Other measures used were the Short Revised Almost Perfect Scale Discrepancy subscale (SAPS-D) [45], the Performance Perfectionism Scale [46], and the OBQ Perfectionism/Certainty subscale [47]. Higher scores on these measures indicate a higher degree of perfectionism.

### 3.4. Dimensions of Perfectionism and Their Relationship with Depressive Symptoms

The search found nine common dimensions of perfectionism discussed in relation to depressive symptoms, as well as some more general perfectionism measures. All but one of these studies were cross-sectional. Cox and Enns [26] conducted the only longitudinal study. This was conducted at two time points one year apart. They did not find a significant change in perfectionism measures with a decrease in depressive symptoms over time.

#### 3.4.1. Concern over Mistakes

CM considers negative reactions to mistakes and often involves interpreting mistakes as failures [42]. Three studies investigated the relationship between CM and depressive symptoms. Two studies found moderate, positive, and significant correlations between CM and depressive symptom measures ranging from *r* = 0.39 to *r* = 0.51 [26,27]. Schrijvers et al. [31] did not find a significant correlation between depressive symptoms and CM. This may be because they used the HDRS, whereas the other studies used the BDI and DPDI. Different scales may assess depressive symptoms differently.

#### 3.4.2. Doubts about Actions

DA is the tendency to doubt one’s actions and believe that projects are never completed to satisfaction [42]. Three studies investigated the relationship between DA and depressive symptoms. Two studies found small–moderate, positive, and significant correlations between DA and depressive symptom measures ranging from *r* = 0.29 to *r* = 0.55 [26,27]. Schrijvers et al. [31] did not find a significant correlation between depressive symptoms and DA; however, this was the only study that did not find a correlation, and it may be because a different scale was used. This study also had a small sample size (39 participants) in comparison to the other two samples, which consisted of 105 and 145 participants, and, therefore, it had low statistical power.

#### 3.4.3. Parental Criticism

PC involves believing that one’s parents are excessively critical [42]. Two studies investigated the relationship between PC and depressive symptom measures. Both found small positive significant correlations, both being *r* = 0.21 [26,27]. In a longitudinal study over a year, Cox and Enns [26] also found a smaller non-significant correlation between PC and depressive symptoms (*r* = 0.13).

#### 3.4.4. Parental Expectations

PE is the tendency to think that one’s parents set very high standards and goals [42]. Two studies investigated the relationship between PE and depressive symptoms. One longitudinal study did not find a significant correlation between PE and measures of depressive symptoms [26]. However, Enns and Cox [27] found small significant correlations between the two factors.

#### 3.4.5. Personal Standards

PS involves setting very high standards for yourself and placing excessive importance on these high standards during self-evaluation [42]. All studies investigating PS (*n* = 3) found no correlation between PS and depressive symptoms [26,27,31]. Cox and Enns [26] did find a slight positive significant correlation at time two (*r* = 0.20); however, this was the only significant finding.

#### 3.4.6. Organisation

Organisation is concerned with the importance of and preference for orderliness [42]. Two studies investigated the relationship between organisation and depressive symptoms and found small negative correlations (*r* = −0.14 and *r* = −012), but these were non-significant [26,27]. Cox and Enns [26] further found no correlation at time point 2.

#### 3.4.7. Self-Oriented Perfectionism

SOP involves setting unrealistic standards for oneself, striving to attain perfection, and critically evaluating oneself when these standards are not met [43]. Six studies investigated the relationship between SOP and depressive symptoms, five of which found significant small–moderate positive correlations, ranging from *r* = 0.135 to *r* = 0.48 [25,26,27,28,29]. Cox and Enns [26] found a non-significant correlation at time one (*r* = 0.11), suggesting that the relationship between SOP and depressive symptoms strengthened over time. However, most of the correlations were small, suggesting that this relationship is weak. Corry et al. [25] did not find a correlation between SOP and the ISS measure of depressive symptoms, suggesting that the correlation also depends on the scale used.

Cheng et al. [23] investigated the relationship between depressive symptoms and positive and negative outcome expectancies of performance due to SOP. They found that the more severe the depressive symptoms, the higher negative (*r* = 0.20) and the lower positive (*r* = −0.28) outcome expectancies an individual portrayed.

#### 3.4.8. Socially Prescribed Perfectionism

SPP involves an individual believing that others have unrealistic standards for them, expect them to be perfect, and criticise them stringently if they do not meet these standards [43]. Five studies investigated the relationship between SPP and depressive symptoms, four of which found moderate correlations ranging from *r* = 0.31 to *r* = 0.49 [25,26,27,28]. Cheng et al. [23] investigated the relationship between depressive symptoms and positive and negative outcome expectancies of performance due to SPP. They found that the more severe the depressive symptoms, the higher negative (*r* = 0.25) and the lower positive (*r* = −0.12) outcome expectancies an individual portrayed.

#### 3.4.9. Other-Oriented Perfectionism

OOP involves setting unrealistic standards for others, placing importance on others being perfect, and evaluating others’ performance harshly [43]. Two studies investigated the relationship between OOP and depressive symptoms. One study found no significant correlation [26]. Enns and Cox [27], however, did find a significant positive correlation. This was small (*r* = 0.20), suggesting that as the severity of depressive symptoms increases, an individual’s expectations and standards of others increase slightly.

#### 3.4.10. Dysfunctional Perfectionistic Beliefs and General Perfectionism

Bahceci et al. [22] used the perfectionism/certainty subscale of the OBQ to investigate obsessive perfectionistic thinking and difficulty tolerating mistakes [47]. They found a significant moderate correlation (*r* = 0.36) between this measure and depressive symptoms. This indicates that as depressive symptom severity increases, obsessive perfectionistic thoughts increase.

### 3.5. Dimensions of Perfectionism and Their Relationship with Manic Symptoms in Mood Disorders

Only four studies looked at the relationship between hypomanic symptoms and levels of perfectionism, all of which were cross-sectional.

#### 3.5.1. Dysfunctional Perfectionistic Beliefs and Maladaptive Perfectionism

Muralidharan et al. [30] found no correlation between manic symptoms and dysfunctional perfectionistic thinking. Meanwhile, Corry et al. [24] investigated the self-critical perfectionism factor of the DAS (DAS-SCP), which involves self-scrutiny and concern over criticism from others. They found a significant and positive correlation between this and manic symptoms, but it was only small (*r* = 0.2). Fletcher et al. [14] used the SAPS-D, which assesses maladaptive perfectionism, perfectionistic concerns, and the discrepancy between an individual’s standards and the extent to which these standards are met. No correlation was found between this and manic symptoms. However, these findings are hard to compare, as the studies used different scales to measure perfectionism and depressive symptoms.

#### 3.5.2. Self-Oriented Perfectionism

Only one study investigated the relationship between SPP and manic symptoms. Corry et al. [25] found that SOP significantly and positively correlated with GBI hypomanic scores (*r* = 0.45) and ISS hypomanic scores (*r* = 0.24).

#### 3.5.3. Socially Prescribed Perfectionism

Only one study investigated the relationship between SPP and manic symptoms. Corry et al. [25] found that SPP significantly and positively correlated with GBI hypomanic scores (*r* = 0.40) and ISS hypomanic scores (*r* = 0.18).

## 4. Meta-Analysis

There was insufficient data for hypomania/mania symptoms. For depression, five studies were included. Data for the remaining seven studies could not be included in the revised meta-analysis due to some perfectionism subtypes only being reported for one study. For example, a correlation between depression and organisation and personal standards was only reported by Schrijvers et al. [31]; therefore, this could not be included. A random effects model analysed the strength of correlations for each individual perfectionism subtype. Figure 2 displays a forest plot of the results. There was a statistically significant relationship, with large effect sizes, between greater depression symptom severity that was positively correlated with more severe levels of concern over mistakes: *r* = 0.40 (CI: *r* = 0.22–0.55), *p* < 0.001 and socially prescribed perfectionism: *r* = 0.38 (CI: *r* = 0.28–0.48), *p* < 0.001. There were small effect sizes but significant correlations between greater depression symptom severity and greater levels of doubting of actions: *r* = 0.26 (CI: *r* = 0.12–0.38), *p* < 0.001; parental criticism *r* = 0.18 (CI: *r* = 0.05–0.30), *p* < 0.01; self-oriented perfectionism *r* = 0.16 (CI: *r* = 0.09–0.22), *p* < 0.001; and other-oriented perfectionism *r* = 0.16 (CI: *r* = 0.03–0.28), *p*< 0.05.

There was no significant correlation between depression symptom severity and perfectionism subtypes of organisation: *r* = 0.02 (CI: *r* = −0.24–0.30), *p* > 0.05 or personal standards: *r* = 0.07 (CI: *r* = −0.06–0.19), *p* > 0.05.

## 5. Discussion

The systematic review and meta-analysis aimed to review and summarise the existing literature on the relationship between mood symptom severity and perfectionism. This systematic review and meta-analysis adds to existing reviews and meta-analyses in the literature on the relationship between psychopathology and perfectionism [11,19,20,21] by showing the relationship between mood symptoms in mood disorders and perfectionism subtypes. The findings of this review are consistent with previous findings that have shown that there is a link between higher levels of perfectionism and more severe depressive symptoms [11,12]. The most investigated and consistent findings were for self-oriented perfectionism and socially prescribed perfectionism. These subtypes, along with concern over mistakes and doubts about actions, had the strongest correlations, although the findings for concern over mistakes and doubts about actions were less consistent. This suggests that these are the most common subtypes of perfectionism in depression. The inconsistencies may be due to different scales being used to assess mood symptoms and varying sample sizes affecting the findings. 

The findings of this meta-analysis further add that there is a differing relationship depending on the perfectionism subtype. Specifically, depression is linked most strongly to concern over mistakes and socially prescribed perfectionism. There is also a less strong but significant relationship for doubting of actions, parental criticism, self-oriented perfectionism, and other-oriented perfectionism. There is no relationship with personal standards and organisation specifically. However, the small number of studies included for each subtype needs to be considered. This may be because organisation and personal standards have previously been categorised as adaptive types of perfectionism, so they correlate less with depressive symptoms, whilst the rest of the subtypes from the Frost Multidimensional Perfectionism Scale (concern over mistakes, doubts about actions, parental criticism, and parental expectations) have been categorised as maladaptive [48]. However, socially oriented and other-oriented perfectionism have also previously been classed as adaptive in some cases [49], so this should be researched further. The small number of studies included for each subtype in our meta-analysis needs to be considered, as this may have impacted the findings.

There is stronger evidence for a relationship with depressive symptoms than manic symptoms in bipolar disorder; however, the limited research in the area meant that a meta-analysis could not be performed for manic symptoms. Manic symptoms involve an individual having inflated self-esteem [1]. It is likely that this would impact levels of perfectionism, as they believe in themselves more in this state. Future research on the link between perfectionism subtypes and manic symptoms is needed to draw conclusions on this.

Whilst all studies included used standardised measures to measure perfectionism and well-validated tools to diagnose disorders, many of the studies used different measures. Most of the studies did not state whether participants were unwell or stable, which may have further impacted the findings, as their responses could have been influenced by their current mood state. All but one study was cross-sectional, making it hard to determine the direction of causality; it may be that perfectionism predisposes to depression worsening over time. It may also be that when depressed, individuals tend to have higher standards for themselves and be more self-critical. This links to a wider debate in the literature about the cognitive mechanisms of depression and bipolar disorder and whether these are stable traits [50] or state mechanisms which are only present during mood episodes [50,51,52]. It has also been suggested that the relationship might work both ways [53], raising the possibility that perfectionism increases vulnerability to later depression, and such perfectionistic beliefs are exacerbated during acute depressive episodes.

This review has clinical implications, as the findings apply to a clinical population of individuals with mood disorders. However, caution should be used when translating these findings because most studies did not state whether the participants were stable or unwell. It suggests that therapies targeting certain maladaptive dimensions of perfectionism may be beneficial for those with mood disorders and may be useful as a target to reduce depression symptom severity. A recent meta-analysis demonstrated the efficacy of cognitive behavioural interventions for perfectionism with medium effect sizes for the symptoms of depression [54]. Some studies have looked at the effectiveness of psychological therapies for the related concept of dysfunctional assumptions in bipolar disorder [55]. However, the systematic review did not identify any studies looking at the effectiveness on perfectionism within bipolar specifically [54]. 

This meta-analysis is limited by its inclusion of only twelve papers overall, and due to data on specific perfectionism subtypes only being reported by some individual studies, there were only five studies in the meta-analysis. Some of the analyses in the meta-analysis were only based on two studies, and, therefore, the results need to be interpreted with caution. The papers included were also not all screened by two independent reviewers, although for a sample of the papers, the agreement was good. Due to the number of papers included, it was also not possible to separate the analyses to see whether there were differences in correlations between unipolar depression and bipolar disorder. 

Future longitudinal studies that assess the relationship between mood severity and perfectionism subtypes over time are warranted, as well as studies focusing on the difference during different phases of the disorder (i.e., euthymic versus depressive and hypomanic states in bipolar disorder) and in particular on the link with (hypo)manic symptoms.

## Figures and Tables

**Figure 1 brainsci-13-00377-f001:**
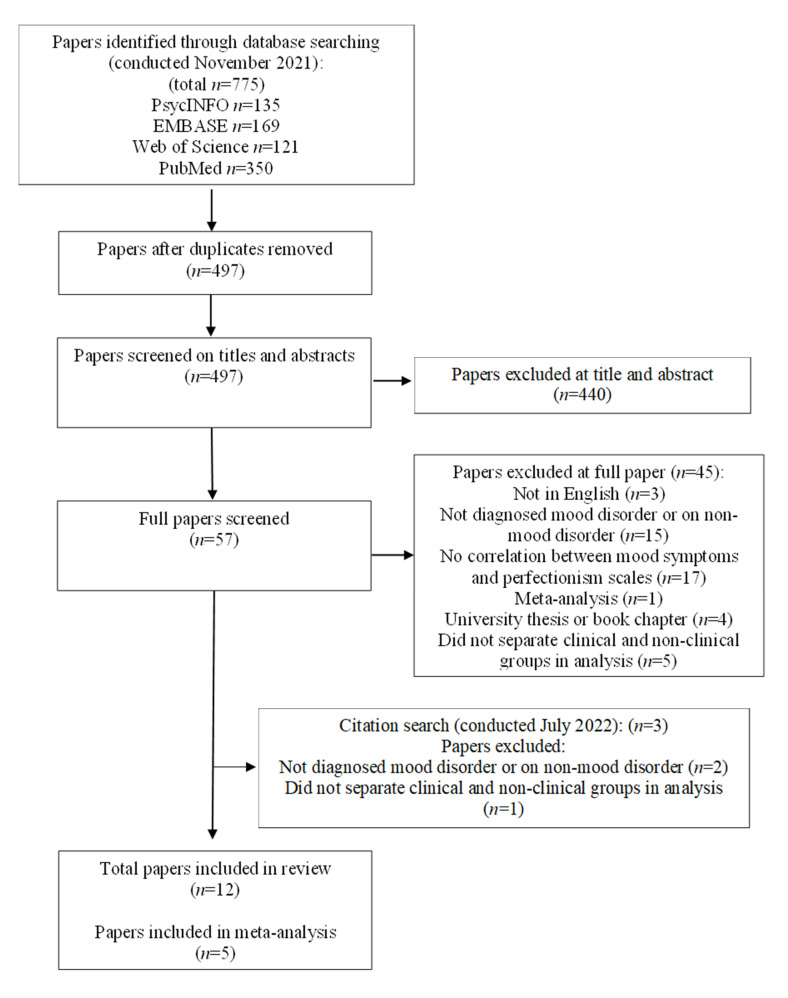
Flow Diagram of the systematic search.

**Figure 2 brainsci-13-00377-f002:**
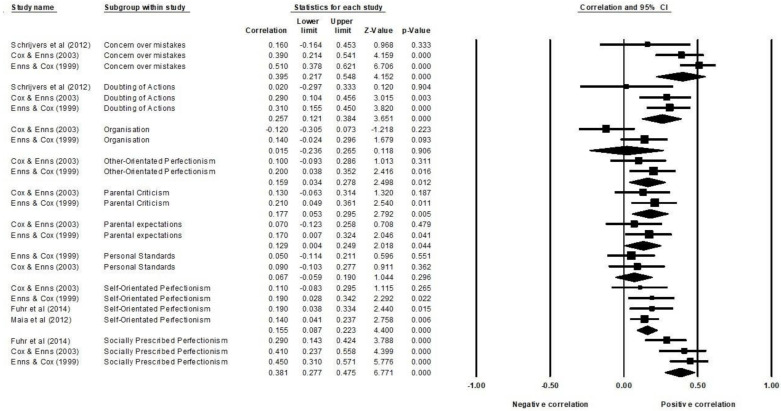
Forest plot of perfectionism subtypes and depression symptom severity [26,27,28,29,31].

**Table 1 brainsci-13-00377-t001:** Study Characteristics and Relevant Findings.

Authors and Date	Country of Study	Study Design	Measures Used	Sample Characteristics		Relevant Findings
				Disorders Studied and Sample Sizes	Age; Recruitment; Stability	
Bahceci et al. (2014) [22]	Turkey	- Cross-sectional - Correlational	- HDRS - OBQ Perfectionism/Certainty subscale	MDD (*n* = 74) Controls (*n* = 74)	Age: MDD *M* = 33.5 Control *M* = 34.8 Recruitment: outpatients Stability: not stated	- Significant positive correlation between the HDRS scores and the perfectionism/certainty subscale in MDD patients (*r* = 0.36, *p* < 0.001).
Cheng et al. (2015) [23]	USA	- Cross-sectional - Correlational	- BDI-II - PPS	MDD (*n* = 49) Controls (*n* = 42)	Age: MDD *M* = 27.38 Control *M* = 25 Recruitment: from internet advertisements Stability: not stated	- Negative socially prescribed performance perfectionism positively and significantly correlated with BDI (*r* = 0.25, *p* < 0.05). - Positive self-oriented performance perfectionism had a slight negative correlation with BDI scores (*r* = −0.28, *p* < 0.01). - Negative self-oriented performance perfectionism negatively correlated with BDI (*r* = 0.20). Not significant. - Positive socially prescribed performance perfectionism had a slight negative correlation with BDI scores (*r* = −0.12). Not significant.
Corry et al. (2013) [24]	Australia	- Cross-sectional - Correlational	- DASS - MADRS - YMRS - DAS self-critical perfectionism factor	Bipolar I disorder (*n* = 97) Bipolar II disorder (*n* = 44)	Age: *M* = 38.56 Recruitment: patients Stability: not stated	- DAS-self-critical perfectionism factor significantly and positively correlated with MADRS (*r* = 0.33, *p* < 0.01) and YMRS (*r* = 0.20, *p* < 0.05).
Corry et al. (2017) [25]	Australia	- Cross-sectional - Correlational - Regression	- ISS - GBI - DASS - HMPS - DAS	MDD (*n* = 105)	Age: *M* = 52.11 Recruitment: from previous BD study Stability: not stated	- SOP significantly and positively correlated with GBI depression (*r* = 0.48, *p* < 0.01) and GBI hypomanic (*r* = 0.45, *p* < 0.01). - SPP significantly and positively correlated with GBI depression (*r* = 0.31, *p* < 0.01) and GBI hypomanic (*r* = 0.40, *p* < 0.01). - SOP had a small, significant, positive correlation with ISS hypomanic (*r* = 0.24, *p* < 0.01), but it did not correlate with (*r* = 0.10). - SPP had a small, significant, positive with ISS hypomanic (*r* = 0.18, *p* < 0.01), but it did not correlate with ISS depressive episode (*r* = −0.10).
Cox & Enns (2003) [26]	Canada	- Longitudinal (questionnaires completed twice, one year apart) - Correlational - Regression	- BDI - HMPS - FMPS	MDD (*n* = 145)	Age: *M* = 41.6 Recruitment: outpatients Stability: not stated	- At time 1, BDI significantly and positively correlated with SPP, (*r* = 0.41, *p* < 0.01), CM (*r* = 0.39, *p* < 0.01), and DA (*r* = 0.29, *p* < 0.01). BDI did not have significant correlations with SOP (*r* = 0.11), OOP (*r* = 0.10), PS (*r* = −0.09), PE (*r* = −0.07), PC (*r* = 0.13), or organisation (*r* = −0.12). - At time 2, BDI significantly and positively correlated with SPP (*r* = 0.49, *p* < 0.01), SOP (*r* = 0.35, *p* < 0.01), CM (*r* = 0.47, *p* < 0.01), PS (*r* = 0.20, *p* < 0.05), PC (*r* = 0.21, *p* < 0.05), and DA (*r* = 0.55, *p* < 0.01). BDI did not have significant correlations with OOP (*r* = 0.12), PE (*r* = 0.15), or organisation (*r* = 0.03).
Enns & Cox (1999) [27]	Canada	- Cross-sectional - Correlational	- DEQ - BDI - HDRS - HMPS - FMPS	BD (*n* = 302): bipolar I (*n* = 242), bipolar II (*n* = 54), bipolar NOS (*n* = 6)	Age: *M* = 43.6 Recruitment: outpatients Stability: not stated	- BDI significantly and positively correlated with SPP (*r* = 0.45, *p* < 0.01), SOP (*r* = 0.19, *p* < 0.05), OOP (*r* = 0.20, *p* < 0.05), CM (*r* = 0.51, *p* < 0.01), DA (*r* = 0.31, *p* < 0.01), PC (*r* = 0.21, *p* < 0.05), and PE (*r* = 0.17, *p* < 0.05). BDI did not significantly correlate with PS (*r* = 0.05) or organisation (*r* = −0.14).
Fletcher et al. (2019) [14]	International (Australia, USA, UK, Canada)	- Secondary data analysis - Cross-sectional - Correlational	- QIDS-SR - MADRS - YMRS - SAPS-D	BD (*n* = 53) MDD (*n* = 58) Controls (*n* = 53) High-risk sample (*n* = 49)	Age: *M* = 44 Recruitment: not stated Stability: not stated	- SAPS-D positively and significantly correlated with QIDS-SR (*r* = 0.31, *p* < 0.01) and MADRS (*r* = 0.24, *p* < 0.01). - SAPS-D did not correlate with YMRS (*r* = 0.07). Not significant.
Fuhr et al. (2014) [28]	Germany	- Cross-sectional - Correlational	- YMRS - HRSD - BDI-II - DAS two subscales: achievement and dependency - HMPS	Postpartum depression (*n* = 386)	Age: *M* = 42.77 Recruitment: outpatients Stability: Stable for 2 months or more	- BDI-II had significant, small correlations with SOP (*r* = 0.19, *p* < 0.05) and SPP (*r* = 0.29, *p* < 0.001). - HRSD had significant, small correlations with SOP (*r* = 0.17, *p* < 0.05) and SPP (*r* = 0.29, *p* < 0.001).
Maia et al. (2012) [29]	Portugal	- Cross-sectional - Correlational - Regression	- BDI-II - HMPS	BD (*n* = 22) Control (*n* = 22)	Age: *M* = 30.08 Recruitment: approached whilst waiting for medical appointment Stability: not stated	- During pregnancy, BDI-II had significant, small, positive correlations with SOP (*r* = 0.14, *p* < 0.01), SPP-Others’ High Standards (*r* = 0.20 *p* < 0.01), and SPP-Conditional Acceptance (*r* = 0.15 *p* < 0.01). - During the postpartum period, BDI-II had significant, small, positive correlations with SPP-Others’ High Standards (*r* = 0.21, *p* < 0.01) and SPP-Conditional Acceptance (*r* = 0.19, *p* < 0.01) but not with SOP (*r* = 0.07).
Muralidharan et al. (2015) [30]	USA	- Cross-sectional - Correlational	- YMRS - HDRS - PANAS - DEQ - DAS-P	MDD (*n* = 39)	Age: BD (*M* = 25.18). Controls (*M* = 26.44) Recruitment: patients Stability: Stabilised for at least 7 days	- No significant correlations between DAS-P and HDRS (*r* = 0.26) or YMRS (*r* = −0.01).
Schrijvers et al. (2010) [31]	Belgium	- Cross-sectional - Correlational	- HDRS - FMPS		*Age: M* = 39 Recruitment: inpatients Stability: on medication	- No significant correlations between HDRS and CM (*r* = 0.16), DA (*r* = −0.02), or PS (*r* = −0.01).

Note. MDD = major depressive disorder; BD = bipolar disorder; HDRS = Hamilton Depression Rating Scale; BDI = Beck Depression Inventory; BDI-II = Beck Depression Inventory II; DASS = Depression, Anxiety, and Stress scale; DASS-21 = Depression Anxiety Stress Scale, 21-Item Version; MADRS = Montgomery Asberg Depression Rating Scale; YMRS = Young Mania Rating Scale; ISS = Internal State Scale; GBI = General Behaviour Inventory; DEQ = Depressive Experiences Questionnaire; QIDS-SR = Quick Inventory of Depressive Symptomatology—Self-Report; PANAS = Positive and Negative Affect Schedule; OBQ = Obsessive Beliefs Questionnaire; PPS = Performance Perfectionism Scale; DAS = Dysfunctional Attitudes Scale; FMPS = Frost Multidimensional Perfectionism Scale; CM = Concern over Mistakes; DA = Doubting of Actions; PE = Parental Expectations; PC = Parental Criticism; PS = Personal Standards; HMPS = Hewitt and Flett Multidimensional Perfectionism Scale; SOP = Self-Oriented Perfectionism; SPP = Socially Prescribed Perfectionism; OOP = Other-Oriented Perfectionism; SAPS-D = Short Revised Almost Perfect Discrepancy Subscale; DAS-P = Dysfunctional Attitudes Scale Perfectionism subscale.

## Data Availability

The comprehensive meta-analysis data file is available upon author request.

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
