# Peer review of "The Relationship between Mood Symptom Severity and Perfectionism Subtypes in Mood Disorders: A Systematic Review and Meta-Analysis"

_brainsci, 2023, doi:10.3390/brainsci13030377_

Round 1

Reviewer 1 Report

1. Details of different scales to measure mood symptoms should be added.

2. Limitations of this study should be discussed.

Author Response

We would like to thank the reviewers for their helpful comments on our paper. We have detailed how we have addressed these below, and we hope that the paper can now be accepted for publication.

We have marked the changes with track changes to make these clear.

Reviewer 1

_____________________________________

  1. Comment: Considering the extremely complex nature of depressive and bipolar disorders, both in terms of pathomechanisms and symptoms, as well as the fact that the boundaries of mental/depressive disorders are still not fully defined, the description of these conditions, presented in lines 51-56, seems to be highly simplistic and general. Please, present the clinical picture of these disorders in more detail.

Response: We have now removed the 3rd paragraph and moved this to the start of the introduction and significantly expanded the overview of depression and bipolar disorder and their impact:

“Depression is a mental illness characterised by symptoms such as  low mood, hopelessness, poor concentration and thoughts about death [1,56], and is estimated to have a lifetime prevalence of 10.8% internationally [34]. Bipolar disorder is a mental illness characterised by episodes of depression with episodes of hypomania or mania characterised by symptoms including increased energy, pressure of speech, grandiose ideas and an increase in goal-directed activity [1,56]. Bipolar disorder has a lifetime prevalence of 1.7% [29] and is linked to a number of negative outcomes including increase suicide risk [15] and reduced life-expectancy [31]. Depression globally is linked to 24.6 million years living with disability per year [55] and substantial health costs [33]. Recent meta-analyses have demonstrated the effectiveness of psychological therapies for both Bipolar Disorder [40] and depression [49].”

____________________________________________

  1. Comment: Taking into account that the article is a review, the introduction section seems to be insufficient in presenting the background for further analyses described in the paper. The Authors claim that “There is a large body of literature on the relationship between psychopathology and perfectionism.”, however, the introduction does not provide the reader with extensive literature data on the subject covered in the article. It is highly recommended to expand this part of the paper with more information/literature reports on e.g. depressive and bipolar disorders, perfectionism, the importance of the subject to therapy, etc.

Response: We have expanded the introduction section by:

  1. Adding more detail into discussion of the Limburg et al. meta-analysis:

However, this meta-analysis mainly looked at psychopathological symptoms over a range of disorders, and it only looked at these in relation to two overarching dimensions of perfectionism: perfectionistic strivings and perfectionistic concerns.

  1. Adding the following section

“Other studies have shown that an increase in perfectionism is correlated with an increase in depression scores [28]. Hence, perfectionism appears to have a predictive relationship with depression; where the trait further contributes to the maintenance of depressive symptoms [26]. Furthermore, treatments targeting perfectionism may be effective in decreasing depressive symptoms [27]. Thus, expanding our understanding of the relationship between mood symptoms and perfectionism subtypes would be beneficial, as this will have implications for the development of efficient clinical treatments. Therapies targeting the reduction of maladaptive perfectionism should be at the forefront of mood disorder treatment.”

  1. Adding another review:

“An earlier review found that perfectionism is higher in anxiety disorders, depression, and eating disorders than in those without, and that perfectionism may explain comorbidity across disorders [17].”

_____________________________________________________

  1. Comment:     The weakest part of the article is the discussion, which, in the reviewer’s opinion, should be the most extended/vast part of this type of publication. In its current form, this section contains only 5 very short paragraphs and refers to 8 (!) publications of other research groups. Unfortunately, it does not provide an extensive comment/discussion of the results of the meta-analysis or the topic. The authors limited themselves to a dozen or so articles, which they described very briefly in the part concerning the results. There is no proper discussion of these observations in light of other articles crucial to the field or a broader commentary on them in this section.

Response: The discussion has been expanded:

  1. To include greater reference to other literature in the field:

‘This systematic review and meta-analysis adds to existing reviews and meta-analyses in the literature on the relationship between psychopathology and perfectionism [17-18,20,35] by showing the relationship between mood symptoms in mood disorders and perfectionism subtypes.’

  1. To discuss the studies included in the review further:

‘The most investigated and consistent findings were for self-oriented perfectionism and socially prescribed perfectionism. These subtypes, along with concern over mistakes and doubts about actions, had the strongest correlations, although the findings for concern over mistakes and doubts about actions were less consistent. This suggests that these are the most common subtypes of perfectionism in depression. The inconsistencies may be due to different scales being used to assess mood symptoms, and varying sample sizes affecting the findings.’

  1. To discuss the lack of findings for manic symptoms further:

‘There is stronger evidence for a relationship with depressive symptoms than manic symptoms in bipolar disorder, however the limited research in the area means that a meta-analysis could not be performed for manic symptoms. Manic symptoms involve an individual having inflated self-esteem [1]. It is likely that this would impact levels of perfectionism as they believe in themselves more in this state. Future research on the link between perfectionism sub-types and manic symptoms is needed to draw conclusions on this.’

  1. To add to the discussion of the meta-analysis findings:

‘This may be as organisation and personal standards have previously been categorised as adaptive types of perfectionism, so they correlate less with depressive symptoms, whilst the rest of the subtypes from the Frost Multidimensional Perfectionism Scale (concern over mistakes, doubts about actions, parental criticism, and parental expectations) have been categorised as maladaptive [54]. However, socially-oriented and other-oriented perfectionism have also previously been classed as adaptive in some cases [6] so this should be researched further.’

______________________________________

  1. Comment: The meta-analysis included in the manuscript covers only 5 articles and, due to how it is presented, brings limited novelty to the subject addressed in the paper. Moreover, the Authors stated that they included 12 articles in the review but only 5 in the meta-analysis. What was the exact reason behind this limitation (as it is not clearly described)?

Response: We have now add to the results:

Data for the remaining seven studies could not be included in the revised meta-analysis due to some perfectionism sub-types only being reported for one study. For example a correlation between depression and organisation and personal standards was only reported by Schrijvers et al. [50], therefore this could not be included

We have also added to the limitations section of the discussion:

This meta-analysis is limited by there only being twelve papers overall, and due to data on specific perfectionism sub-types only being reported by some individuals studies, there were only five studies in the meta-analysis. ______________________________________

 Minor concerns:

  1. Comment: Please, read the text carefully and correct minor misspellings, language, grammar and editorial errors, e.g.: line 11 “ This study aimed to see if different types of perfectionism is linked…”; line 73 “This study therefore aimed to conducted…”; in the discussion “The systematic review and meta-analysis aimed to to review and…”; again in the discussion “There there is no relationship with personal…”; at the very end “This research received not funding.”. Line 15: the phrase „Five of these papers” seems to be accidentally/incorrectly placed.

Response: The text has been proofread and all spelling errors, grammatical errors, and editorial errors have been corrected.

  1. Comment: Please provide a reference(s) to support the statement in line 33.

Response: The paper ‘Limburg, K.; Watson, H. J.; Hagger, M. S.; Egan, S. J. The relationship between perfectionism and psychopathology: A meta-analysis. J Clin Psychol 2016, 73, 1301–1326, doi.org/10.1002/jclp.22435.’ has been referenced here to support this statement.

______________________________

  1. Comment:It seems to be confusing if the Authors aimed to examine the correlation between different types of perfectionism (“perfectionism subtypes”) and depression (line 11) or different types of depression and perfectionism (line 72). This is particularly misleading when comparing statements in lines 71-75. Please, correct this issue within the article as the aim of the study should be clearly stated/transparent for the reader.

Response: Line 72 has been corrected to “There have also been no meta-analyses which have examined whether different subtypes of perfectionism have a different relationship with depression.” from “There have also been no meta-analyses which have examined whether different types of depression have a different relationship with perfectionism.”

______________________________

  1. Comment: The sentence “Whilst the DAS-P does not directly measure a dimension of perfectionism, it is within the domain of perfectionism as it looks at dysfunctional perfectionistic thinking.” from lines 103-104 is repeated in identical form in lines 116-117.

Response: We have removed both of these sentences as we did not feel they were necessary.

_________________________

  1. Comment: The references section needs editing/corrections.

Response: The formatting of the reference section and the citations have been corrected according to the MDPI referencing guidelines.

Reviewer 2 Report

The manuscript by Lea et al. aims to investigate the correlation between sub-types of perfectionism and depression or bipolar disorder. The authors sought to include in the article a literature review and a meta-analysis.

Although the subject of the article is highly interesting and could bring important information about perfectionism in the context of mental disorders, the manuscript in its present form contains some flaws and concerns that must be addressed.

Major concerns:

1.      Considering the extremely complex nature of depressive and bipolar disorders, both in terms of pathomechanisms and symptoms, as well as the fact that the boundaries of mental/depressive disorders are still not fully defined, the description of these conditions, presented in lines 51-56, seems to be highly simplistic and general. Please, present the clinical picture of these disorders in more detail.

2.      Taking into account that the article is a review, the introduction section seems to be insufficient in presenting the background for further analyses described in the paper. The Authors claim that “There is a large body of literature on the relationship between psychopathology and perfectionism.”, however, the introduction does not provide the reader with extensive literature data on the subject covered in the article. It is highly recommended to expand this part of the paper with more information/literature reports on e.g. depressive and bipolar disorders, perfectionism, the importance of the subject to therapy, etc.

3.      The weakest part of the article is the discussion, which, in the reviewer’s opinion, should be the most extended/vast part of this type of publication. In its current form, this section contains only 5 very short paragraphs and refers to 8 (!) publications of other research groups. Unfortunately, it does not provide an extensive comment/discussion of the results of the meta-analysis or the topic. The authors limited themselves to a dozen or so articles, which they described very briefly in the part concerning the results. There is no proper discussion of these observations in light of other articles crucial to the field or a broader commentary on them in this section.

4.      The meta-analysis included in the manuscript covers only 5 articles and, due to how it is presented, brings limited novelty to the subject addressed in the paper. Moreover, the Authors stated that they included 12 articles in the review but only 5 in the meta-analysis. What was the exact reason behind this limitation (as it is not clearly described)?

Minor concerns:

1.      Please, read the text carefully and correct minor misspellings, language, grammar and editorial errors, e.g.: line 11 “ This study aimed to see if different types of perfectionism is linked…”; line 73 “This study therefore aimed to conducted…”; in the discussion “The systematic review and meta-analysis aimed to to review and…”; again in the discussion “There there is no relationship with personal…”; at the very end “This research received not funding.”. Line 15: the phrase „Five of these papers” seems to be accidentally/incorrectly placed.

2.      Please provide a reference(s) to support the statement in line 33.

3.      It seems to be confusing if the Authors aimed to examine the correlation between different types of perfectionism (“perfectionism subtypes”) and depression (line 11) or different types of depression and perfectionism (line 72). This is particularly misleading when comparing statements in lines 71-75. Please, correct this issue within the article as the aim of the study should be clearly stated/transparent for the reader.

4.      The sentence “Whilst the DAS-P does not directly measure a dimension of perfectionism, it is within the domain of perfectionism as it looks at dysfunctional perfectionistic thinking.” from lines 103-104 is repeated in identical form in lines 116-117.

5.      The references section needs editing/corrections.

Author Response

We would like to thank the reviewers for their helpful comments on our paper. We have detailed how we have addressed these below, and we hope that the paper can now be accepted for publication.

We have marked the changes with track changes to make these clear.

Reviewer 2

  1. Comment: Details of different scales to measure mood symptoms should be added.

Response: We have now added more detail to this section:

The most commonly used measures of depressive symptoms were the Beck Depression Inventory (BDI; Beck et al., 1996)[5], a 21-item self-report measure of depression symptoms with items rated from 0 to 3 for example “I do not feel sad” (0) up to “I am so sad and unhappy all the time that I can’t stand it” (3).  The Hamilton Depression Rating Scale (HDRS; Hamilton, 1960) [24] is a clinician-rated measure of 17 items rating severity of symptoms such as work and activities (No difficulty=0, stopped working=4, and feelings of guilt (absent=0. up to hears accusatory or denunciatory voices or threatening visual hallucinations=4). The 21-Item Version of the Depression Anxiety Stress Scale (DASS-21; Lovibond & Lovibond, 1995) [37] depression subscale has seven self-reported items to measure symptom severity via questions such as “I couldn’t seem to experience any positive feeling at all” rated from 0 (did not apply to me at all) up to 3 (applied to me very much or all of the time). , and tThe Depressive Experiences Questionnaire [7](Blatt et al., 1976) was also used, this is a self-report measure of experiences which are commonly seen in depression but not core depression symptoms such as helplessness, guilt, and difficulties in relationships with family members. Other, less frequently used measures were the Quick Inventory of Depressive Symptomatology-Self-Report [48](Rush et al., 2003), a 16-item self report measure with questions about symptoms over the past week such as falling asleep (never taking longer than 30 minutes to fall asleep up to talking more than 60 minutes most of the time). The Montgomery-Asberg Depression Rating Scale [41](Montgomery & Asberg, 1979) is a clinician rated rating to assess symptom severity of 10 items such as difficulty concentrating and difficulty starting and completing activities. The Positive and Negative Affect Schedule [13](Crawford & Henry, 2004), this includes a 10 item scale on negative affect, completed via self report for the past week on a scale from not at all (1) to very much (5) for items such as “ashamed” and “interested”.

The most frequently used measure for manic symptoms was the Young Mania Rating Scale, an 11 item clinician administered scale which assesses the severity of (hypo)manic symptoms, such as energy levels, grandiosity, sleep, and irritability, over the past 48 hours [57]. The Internal State Scale (ISS), a 16 item self-report scale, was used to assess both depressive and manic symptoms with statements such as “Today I feel impulsive” and “Today I feel depressed”; these are rated from 1 (Not at all/Rarely) to 100 (Very much so/Much of the time) [4]. The General Behaviour Inventory (GBI), a 73 item self-report scale, was also used to assess both depressive and manic symptoms using questions such as “Has your mood or energy shifted rapidly back and forth from happy to sad or high to low?” which are rated from 0 (Never or hardly ever) to 3 (Very often/Almost constantly) [14]. On all these measures, higher scores indicate more severe mood symptoms.

  1. Comment: Limitations of this study should be discussed.

Response: We have now added to the discussion section a paragraph on the limitations:

This meta-analysis is limited only including twelve papers overall, and due to data on specific perfectionism sub-types only being reported by some individuals studies, there were only five studies in the meta-analysis. Some of the analyses in the meta analysis were only based on two studies and therefore the results need to be interpreted with caution. The papers included were also not all screened by two independent reviewers, although for a sample of the papers the agreement was good. Due to the number of papers included, it was also not possible to separate the analyses to see if there were differences in correlations between unipolar depression and bipolar disorder.